# Scalable Lightweight Protocol for Interoperable Public Blockchain-Based Supply Chain Ownership Management

**DOI:** 10.3390/s23073433

**Published:** 2023-03-24

**Authors:** Jing Huey Khor, Michail Sidorov, Seri Aathira Balqis Zulqarnain

**Affiliations:** 1Department of Electrical and Electronic Engineering, University of Southampton Malaysia, Iskandar Puteri 79100, Malaysia; 2Department of Computer Science (IDI), Norwegian University of Science and Technology, 7034 Trondheim, Norway; 3School of Electronics and Computer Science, University of Southampton, Southampton SO17 IBJ, UK

**Keywords:** blockchain, interoperability, IoT, scalable protocol

## Abstract

Scalability prevents public blockchains from being widely adopted for Internet of Things (IoT) applications such as supply chain management. Several existing solutions focus on increasing the transaction count, but none of them address scalability challenges introduced by resource-constrained IoT device integration with these blockchains, especially for the purpose of supply chain ownership management. Thus, this paper solves the issue by proposing a scalable public blockchain-based protocol for the interoperable ownership transfer of tagged goods, suitable for use with resource-constrained IoT devices such as widely used Radio Frequency Identification (RFID) tags. The use of a public blockchain is crucial for the proposed solution as it is essential to enable transparent ownership data transfer, guarantee data integrity, and provide on-chain data required for the protocol. A decentralized web application developed using the Ethereum blockchain and an InterPlanetary File System is used to prove the validity of the proposed lightweight protocol. A detailed security analysis is conducted to verify that the proposed lightweight protocol is secure from key disclosure, replay, man-in-the-middle, de-synchronization, and tracking attacks. The proposed scalable protocol is proven to support secure data transfer among resource-constrained RFID tags while being cost-effective at the same time.

## 1. Introduction

Scalability is the main concern for public blockchain-based Internet of Things (IoT) applications. As the number of IoT devices increases each year, coupled with the emergence of the 5G network, this trend is expected to accelerate, driven by the release of a new generation of IoT devices. Existing public blockchains try to solve scalability issues by using different approaches such as on-chain (e.g., sharding) and off-chain (e.g., using sidechain, layer-2 scaling) methods, or by completely using different data structures (e.g., Direct Acyclic Graph used in IOTA), as discussed in [1]. Ethereum 2.0, currently known as the Consensus Layer, is expected to offer a throughput of up to 100,000 transactions per second (TPS) when sharding is implemented. One of its existing technologies—rollups—is already used to improve scalability through layer-2 protocols [2]. This is achieved by offloading heavy computational processes from MainNet to a rollup-specific chain which, in turn, speeds up the transactions. Two types of rollups were introduced by the Ethereum 2.0 blockchain: optimistic rollups and zk-rollups. The former rollups have a long transaction finality time due to their fraud-proof mechanism, which is used to detect incorrectly calculated transactions. In contrast, zk-rollups offer fast finality but require heavy computation for the proving system (e.g., Zk-SNARK) [3]. Both of these rollup solutions have their own weaknesses; therefore, instead of improving them, there is a need for an alternative scaling solution that can be supported by IoT devices. There are several challenges, including the use of resource-constrained IoT devices themselves. Typically, supply chain management solutions use proprietary low-cost Radio Frequency Identification (RFID) tags for goods tracking and ownership management. These tags are classified as Class I IoT devices, meaning they have limited resources and processing capabilities to support complex cryptographic algorithms [1]. In order to enable more transparent, secure, and efficient supply chain management of RFID-tagged goods, this paper proposes a scalable protocol that allows for a secure batch ownership transfer of these tagged goods using an Ethereum public blockchain. While Ethereum 2.0 with rollups significantly reduces overall transaction fees, the proposed solution can further decrease them, as only a single fee is charged for managing a batch of IoT devices, ensuring increased scalability. The proposed scalable protocol is designed using a lightweight cryptographic algorithm to protect resource-limited IoT devices from common security attacks. These security attacks include key disclosure, replay, man-in-the-middle, de-synchronization, and tracking attacks [4]. The protocol is scalable with the help of an InterPlanetary File System (IPFS), and is integrated with a public blockchain to perform transparent IoT device ownership data transfer in batches using only a single transaction. The integration of a public blockchain with this proposed solution is essential for transparent ownership data transfer, data integrity, and access to on-chain data. The main contributions of this paper are as follows:1.A novel scalable public blockchain-based lightweight protocol using bitwise exclusive-OR and simple permutation operations is presented. Its purpose is to perform a secure batch ownership data transfer associated with resource-limited RFID tags;2.The proposed scalable lightweight protocol is able to protect RFID-tagged goods in the supply chain system from key disclosure, replay, man-in-the-middle, de-synchronization, and tracking attacks;3.The proposed lightweight protocol offers partial transparency for interoperable supply chains wherein the public can only view transaction records. Only legitimate owners are allowed to view the full supply chain details through the IPFS;4.The proposed protocol allows offline data transfer in batches, which further reduces transaction costs.

The remainder of the paper is organized as follows: Section 2 describes related work. Section 3 presents the designed scalable lightweight protocol together with a proof of concept in Section 4. Section 5 and Section 6 demonstrate the theoretical and formal analysis of the designed protocol. Section 7 analyzes the performance of the proposed protocol, and Section 8 concludes the paper.

## 2. Related Work

Public blockchains specifically designed for tracking supply chain systems, such as Vechain and Waltonchain, tend to have lower throughput and higher transaction fees compared to other high-performance public blockchains that are not focused on supply chain systems, such as Polygon, IOTA, and Solana. The average throughput of the VeChain network is 165 TPS [5]. In contrast, the Waltonchain network throughput is approximately 13.5 TPS [6]. The network latency, i.e., block generation time in this case, of VeChain is 10 s and around 30.73 s for Waltonchain. These network latencies are quite high compared to the high-performance public blockchains, as shown in Table 1. VeChain uses a dual-token system to prevent fees from being volatile. In April 2021, these fees were reduced to 1x10^13^ Wei (equivalent to USD 0.027 per transaction up to the time of writing) to attract enterprise interest in using this blockchain [5]. Unlike VeChain, Waltonchain’s transaction costs fluctuate similarly to fees in the Ethereum network.

Although high-performance public blockchains offer high throughput and relatively low transaction fees, as shown in Table 1, they have not been widely used in resource-constrained and data-sensitive IoT applications. Several research works have been presented to improve public blockchain scalability by proposing scalable storage models [7,8], cross-chain integration protocols [9], and efficient consensus protocols [10,11,12]. Off-chain data storage can improve blockchain scalability as on-chain data introduces high computational and resource overheads. A blockchain storage model was designed by Chen et al. using a Distributed Hash Table (DHT) and an IPFS to improve its scalability [7]. An IPFS is a distributed file storage protocol that has been used to enable peer-to-peer file sharing for a variety of IoT applications, such as healthcare [8] and supply chains [13]. An IPFS is used not only as off-chain storage to enhance blockchain storage or store sensitive data but also as a tool to achieve interoperability [14,15]. Sidechains can also be used to achieve scalability and interoperability. For example, Rozman et al. proposed a cross-chain integration protocol to create a scalable framework for the Ethereum blockchain and the Xdai sidechain network to enable shared manufacturing [9]. However, in order to enable efficient and secure cross-chain transfers between the blockchain’s main chain and a sidechain, more work is needed [16].

Although efficient data storage models and cross-chain integration protocols can increase blockchain data throughput, the scalability of blockchains can be significantly improved using lightweight consensus algorithms. The first-generation blockchain (e.g., Bitcoin) and most of the second-generation blockchains (e.g., Ethereum 1.0, Litecoin, Monero, etc.) use the Proof-of-Work (PoW) algorithm to achieve consensus. However, blockchains that use PoW have low throughput; thus, other consensus algorithms, such as Proof of Stake (PoS), have been used by newer blockchain generations for better scalability. Some public blockchains provide high scalability with high throughput using scalable consensus algorithms, such as Polygon, which uses PoS, or Solana, which uses Proof of History (PoH). These blockchains have been widely integrated with Ethereum applications to provide higher throughput and lower transaction fees, as shown in Table 1. Some researchers proposed new consensus algorithms to increase blockchain throughput and reduce communication overhead, such as dynamic PoW [10], Zyzzyva consensus protocol [11], and the Groupchain consensus protocol [12]. 

To summarize, public blockchains designed for tracking goods along supply chain systems, i.e., Vechain and Waltonchain, have lower throughput and higher transaction fees compared to high-performance public blockchains that are not focused on supply chains, such as Polygon, IOTA, or Solana. Several approaches, such as off-chain data storage, cross-chain integration protocols, and efficient consensus protocols, have been proposed to improve public blockchain scalability. Additionally, the use of PoS or PoH lightweight consensus algorithms and newly designed consensus protocols can significantly improve public blockchain scalability. However, even though the aforementioned solutions can improve throughput rates, none of them propose a scalable and interoperable supply chain solution for resource-constrained IoT devices. Thus, this paper addresses this research gap by proposing a scalable lightweight protocol for public blockchain-based interoperable supply chain systems that use resource-limited IoT devices.

**Table 1 sensors-23-03433-t001:** Performance Comparison Among Public Blockchains.

Description	Polygon	IOTA	Solana	Vechain	Waltonchain
Throughput (TPS)	7000	1000	50,000	165	13.5
Network latency (s)	2.5	12.00	3.41	10	30.73
Transaction fees (USD)	0.00002	Free	0.00001	0.027	Fluctuate

All data have been compiled from [5,6,17,18,19].

## 3. Scalable Lightweight Protocol for Public Blockchain-Enabled Supply Chains

### 3.1. Lightweight Permutation Operation

Several proposed solutions use the concept of permutation to enhance the security of RFID protocols [20,21,22]. However, [20] requires heavy computations that are unsuitable for resource-constrained passive RFID tags. In [21,22], the permutation operations must analyze every bit in a string to rearrange another string. These operations have a complexity of O(n) according to Big O notation. A lightweight permutation operation is introduced in this paper that analyzes 3 characters for a 64-character hexadecimal string. This operation removes a certain number of characters from a string and inserts them into a specific position, as shown in Figure 1. Each of these three indexing operations has a complexity of O(1) only. Suppose there are two 256-bit *A* and *B* strings (two hexadecimal strings of length 64). A newly proposed lightweight permutation operation *Per(A,B)* refers to a process in which a certain length of characters from string *A* are removed from either the left- or right-hand side based on string *B*, and then inserted into a specific position of string *A*, as described in Table 2. 

### 3.2. Scalable Lightweight Protocol

Ethereum blockchain was used in the proof of concept stage of this work since it supports smart contracts at its core. This further allowed creating a decentralized web application for the proposed supply chain system that interacts with these smart contracts. The current system involves five parties: tag, reader, supply chain node, public blockchain, and IPFS. Basic supply chain nodes consist of manufacturers, distributors, retailers, and end-users. The functionality of supply chain nodes is described in Table 3. The proposed protocol is designed using bitwise exclusive-OR and lightweight permutation operations that allow it to be used with resource-constrained low-cost RFID tags. In contrast, resource-rich supply chain nodes are designed to perform heavier computations, such as hashing Content Identifiers (CID) using the SHA-256 function. In addition, the supply chain node is required to perform the Elliptic Curve Integrated Encryption Scheme using the secp256k1 curve to encrypt or decrypt CIDs and files uploaded to the IPFS. An assumption is made that the communication channel between supply chain nodes is secure, whereas the communication channel between the reader and a tag is insecure. Notations used in the proposed protocol are described in Table 4.

The proposed scalable lightweight protocol consists of two phases: the initial phase and the authentication phase. The initial phase involves steps that need to be performed before the authentication phase can take place. These steps are as follows:1.Each supply chain node creates an account in the Ethereum blockchain and obtains a public key and a private key;2.The manufacturer node generates *ID* and *K* for each RFID tag and stores the *ID* and *K* pairs, and two generated random numbers, *n* and *q*, in a text file called the *DATA* file;3.The manufacturer node hashes *q||n* to compute *Hq*. It then stores the *Hq* string as well as *ID* and *K* pair in each RFID tag;4.The manufacturer node encrypts the *DATA* file with its public key and uploads the encrypted *DATA* file to the IPFS. A *CID*, *Qm*, generated by the IPFS is returned to the manufacturer node;5.The manufacturer supply chain node encrypts the *Qm* with its public key to obtain *H*. A transaction is then made using the Ethereum blockchain, where the *H* string is included as input data in the transaction. A transaction hash, *Tx*, is generated once the transaction is completed;6.The manufacturer supply chain node can include supply chain data to be shared with other supply chain nodes in the *DATA* file.

The authentication phase is conducted when RFID tags communicate with the supply chain node’s reader. Data transfer happens during the authentication phase when the *DATA* file is encrypted with the next supply chain node’s public key. In order to prevent the old owner from tracking the RFID transaction, the *DATA* file is encrypted using the current supply chain node’s public key as listed in steps 5–9 below. In this case, no ownership data transfer happens. The description of the proposed lightweight protocol shown in Figure 2 is as follows:

**Figure 2 sensors-23-03433-f002:**
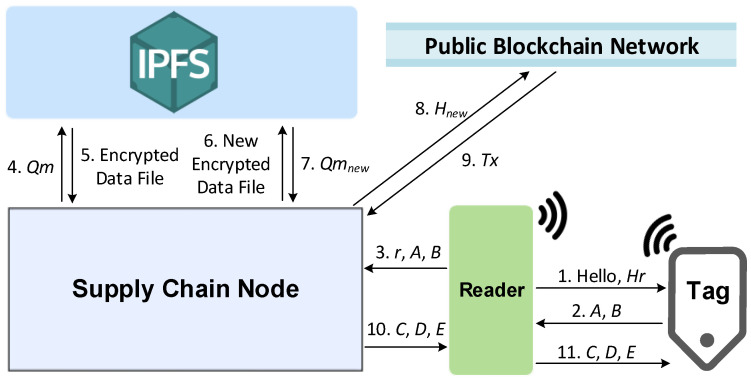
Authentication phase of the proposed lightweight protocol.

1.The reader generates a random number, *r*, and computes *Hr* to initiate a session by hashing the *r* value. It then sends a *Hello* message and the *Hr* value to the tag;2.After receiving both of the messages, the tag generates a random number *m*. The tag then computes *ID_F_* and *K_F_* from its stored *ID* and *K*. Next, the tag computes messages *A* and *B* using *ID_F_*, *K_F_*, and *Hq* values, as well as the *m* and *Hr* values. The tag sends the computed messages, *A* and *B*, to the reader;
*ID_F_* = *Per(ID*,*Hr)*(1)
*K_F_* = *Per(K*,*Hr)*(2)
*A* = *Hq*⊕*ID_F_* ⊕*m*⊕*Hr*(3)
*B* = *Hq*⊕*K_F_*⊕*m*(4)3.The reader forwards *r* and messages *A* and *B* to the supply chain node X (i.e., manufacturer supply chain node);4.The supply chain node X obtains the information using the latest *H* value stored on the blockchain from the decentralized web application. It then decrypts the *H* value to obtain its *Qm* value using its private key;5.The supply chain node X obtains the encrypted *DATA* file from the IPFS. Next, it decrypts the file and computes *Hq*′ by hashing *q||n*, where *q* and *n* are obtained from the file. In order to obtain the correct *ID* and *K* pair, the supply chain node X needs to compute *ID_F_*′ and *K_F_*′ using all *ID* and *K* values stored in the *DATA* file. It then extracts *m*′ from message *B* using the computed *Hq*′ and *K_F_*′ values. It then computes *A*′ using the computed *Hq*′, the hash value of *r*, the extracted *m*′ value, and the *ID_F_*′ value paired with the *K_F_*′ value. If the computed *A*′ is equal to the received *A*, then the correct *ID* and *K* pair is found in the *DATA* file. Otherwise, the current session is terminated;
*m*′= *B*⊕*Hq*′⊕*K_F_*′(5)
*A*′ = *Hq*′⊕*ID_F_*′⊕*m*′⊕*Hr*(6)6.The supply chain node X generates a random number, *s*. It then updates the *ID* and *K* pair and stores them in the *DATA* file. In addition, the random value *r* is also added to the *DATA* file as a new *n* value. The node then computes *Hq_new_* by hashing *q||r*.
*ID_new_* = *ID*⊕*m*⊕*s*(7)
*K_new_* = *K*⊕*m*⊕s(8)
Supply chain node X encrypts the *DATA* file with the public key of supply chain node Y (i.e., distributor supply chain node), and uploads the encrypted *DATA* file to the IPFS.7.IPFS generates an IPFS *CID*, *Qm_new_*, and returns it to the supply chain node X;8.Supply chain node X encrypts the *Qm_new_* with the public key of supply chain node Y to obtain *H_new_*;9.Supply chain node X performs a transaction on the Ethereum blockchain and sends the transaction with the *H_new_* string as input data to supply chain node Y;10.The supply chain node X computes *ID_S_* and *K_S_*. Next, it computes messages *C*, *D,* and *E*, and sends the messages to the reader;
*ID_S_* = *Per(ID*, *Hq_new_)*(9)
*K_S_* = *Per(K*, *Hq_new_)*(10)
*C* = *Hq_new_*⊕*m*(11)
*D* = *K_S_*⊕*s*(12)
*E* = *Hq_new_*⊕*ID_S_*⊕*s*(13)11.The reader forwards messages *C*, *D*, and *E* to the tag. After receiving the messages, the tag extracts *Hq_new_*′ from message *C* using its *m* value. Based on its computed *Hq_new_*′ value, it then computes *ID_S_*′ and *K_S_*′. It extracts *s*′ from message *D* using its computed *K_S_*′ value. The tag then computes *E*′ using the extracted *Hq_new_*′ and *s*′, together with its computed *ID_S_*′ value. If the computed *E*′ is equal to the received *E*, then the tag proceeds with updating its *ID* and *K* pair as well as its *Hq* string. Otherwise, the session is terminated.
*Hq_new_*′ = *C*′⊕*m*(14)
*s*′ = *D*′⊕*Ks*′(15)

Supply chain nodes, such as manufacturers, distributors, and retailers, often would prefer to perform multiple tag ownership transfers since a large number of RFID tags are involved in the process. In a single tag transfer, i.e., when a tagged object is transferred from the retailer supply chain node to the end-user node, a single tag *K* and *ID* are stored in the *DATA* file and encrypted with the end-user’s public key. In contrast, multiple tag *K* and *ID* values are stored in the *DATA* file and encrypted using the supply chain node’s public key for batch transfers. Since all of the tags in a batch use the same q and r values, they have the same *Hq_new_* value. After the reader has finished reading all of the tags, the *DATA* file is encrypted and uploaded to the IPFS. A new *Qm* returned from the IPFS is encrypted and stored on the blockchain to enable the supply chain node to retrieve the encrypted *DATA* file during the next authentication phase.

## 4. Proof of Concept

A proof of concept for the proposed scalable lightweight protocol was developed and deployed on the Ethereum Goerli testnet. The smart contract was written in Solidity. Its purpose is to manage RFID tag ownership transfer in a supply chain system. The smart contract consists of one external *setValue()* and one public custom-defined node structure. The *setValue()* is a mutator function to store the node *ID* (unsigned integer data type) and the encrypted *CID*, also known as *H* (byte array data type), as shown in Figure 3. 

The decentralized web application shown in Figure 4 consists of three sections described below:1.Ownership Transfer:
(a)File encryption: the *uploadipfs()* function is called to encrypt the *DATA* file and upload it to the IPFS. The resultant *CID* generated from the IPFS is encrypted to obtain *H*;(b)Ownership transfer: A transaction is made using the *setValue()* function. The supply chain node *ID* and the *H* values are sent as input in the transaction to a designated address.2.View Transaction: the *getvalue()* function is called where data, including timestamp, sender and receiver addresses, and *H*, are obtained using Etherscan Ethereum Developers Application Programming Interfaces (APIs);3.Retrieve File: the *getfile()* function is called to decrypt *H* and obtain the *CID* plaintext, *Qm*. The uploaded encrypted *DATA* file is downloaded from the IPFS based on the *CID* value. The encrypted *DATA* file is then decrypted using a private key.

## 5. Security Analysis of the Proposed Protocol

The security of the proposed protocol was analyzed against five attacks. Certain assumptions are made based on the Dolev–Yao intruder model to aid in this analysis, described as follows:1.It is possible for the attacker to initialize the communication both with the tag and the reader;2.It is possible for the attacker to eavesdrop, block, and modify the messages sent during the communication sequence between the tag and the reader;3.An attacker is unable to obtain the private key of asymmetric cryptography for each supply chain node.

### 5.1. Key Disclosure Attack

An attacker is unable to retrieve the secret information, *ID*, and *K* pairs from the *DATA* file because asymmetric encryption is used to encrypt the file. The same applies to the IPFS *CID*, where *Qm* is also encrypted with the supply chain node’s public key to prevent the attacker from obtaining the encrypted *DATA* file in the IPFS. This is an additional level of protection for secret *ID* and *K* pairs. A random number *m* is used to encrypt the permutated *ID* and *K* values for each new session in a communication channel between the tag and the reader. This random number is not sent in plaintext, as explained previously. In order to limit the number of random guesses, the attacker has to identify the value of *m*; a threshold of three attempts is set. The reader terminates the session in case this threshold is exceeded, and a new session is initialized with a new random number *r*. The tag then has to use the hash value of *r* and the newly generated random number, *m*, to compute messages *A* and *B*.

The attacker cannot perform brute-force attacks to obtain *ID* and *K* values due to the limited number of trials. Furthermore, guessing the values of *ID* and *K* solely from messages *C*, *D*, and *E* is out of the question since random numbers *s*, *m*, and *Hq_new_* are used to encrypt these messages, respectively. The *Hq_new_* number is different for each session since it is computed by hashing a concatenation of *q* and a random number, *r*. In addition, *ID* and *K* are updated for each new successful session. This increases the difficulty for the attacker to obtain the *ID* and *K* values.

### 5.2. Replay Attack

An attacker may try to perform a replay attack. The process typically involves capturing and delaying messages, *A* and *B* or *C*, *D*, and *E* in this case, and then fraudulently replaying them to the reader. Two scenarios are possible, described below; however, they are ineffective. 

1.The attacker captures messages *A* and *B* and replays them to the reader at the next session. Since the messages are encrypted with new random numbers *Hr* and *m* for every new session, the supply chain node is not be able to authenticate them; 2.The attacker captures messages *C*, *D*, and *E* and replays them to the tag during the next session. The tag is unable to authenticate the messages because they are encrypted with different *Hq_new_*, *m*, and *s* random numbers for each new session.

The attacker fails to convince both the reader and the tag to authenticate the replayed messages based on the above. Therefore, the protocol is resistant to replay attacks.

### 5.3. Man-in-the-Middle Attack

Man-in-the-middle (MITM) attacks are eavesdropping attacks and are accomplished by the adversary inserting themselves between the tag and the reader in order to impersonate both parties. The following scenarios demonstrate that the attack would be unsuccessful: 1.For example, an attacker captures messages *A* and *B* and blocks them from being sent to the reader. These messages are modified and only then sent to the reader. However, since the attacker was unable to obtain the correct values of *K*, *ID*, *m*, and *Hq*, the supply chain node is unable to authenticate the modified messages; 2.An attacker captures messages *C*, *D*, and *E* and blocks them from being sent to the tag. These messages are modified and only then sent to the tag. However, the original messages were encrypted with *m*, *s*, and *Hq_new_*, respectively, which are unknown to the attacker. Therefore, the tag is unable to authenticate the modified messages.

Based on the above, the proposed protocol is considered to be secure from man-in-the-middle attacks.

### 5.4. De-Synchronization Attack

A de-synchronization attack is a type of an attack wherein the attacker tries to break synchronization between the tag and the reader. Several scenarios are possible; however, all of these are ineffective since RFID tags and readers can still communicate during the following sessions either using the current or previous versions of stored values. For example:1.The attacker interferes and blocks messages *A* and *B* from reaching the reader. In this scenario, the reader keeps waiting for messages from the tag. If the messages are not received, the current session is terminated after a certain period of time;2.The attacker blocks messages *C*, *D*, and *E* from reaching the tag. Thus, the tag is unable to update its data, including the *ID* and *K* pair, and the *Hq* string. As a result, the *ID* and *K* values stored in the tag differ from those stored in the latest *DATA* file uploaded by the supply chain node. The supply chain node can obtain the previously encrypted *DATA* file using the previous *H* string obtained from the decentralized web application. The supply chain node proceeds with steps 5–11 in order to confirm the *Hq*, *ID_S_*, and *K* are synchronous between the tag and supply chain node. The same processes apply to attacks that happen during data transfer between supply chain node X and supply chain node Y.

This method, unlike other state-of-the-art solutions, allows secret data to be re-synchronized without sending back RFID-tagged goods to the old owner.

### 5.5. Tracking Attack

This type of attack is typically used for unauthorized tracking of RFID-tagged goods. We considered several scenarios that show that the attack is ineffective.

Attackers might eavesdrop on a session and obtain *K_F_*⊕*ID_F_*⊕*Hr* by performing *A*⊕*B*. They can then obtain *K_F_*⊕*ID_F_* by performing *K_F_*⊕*ID_F_*⊕*Hr* with *Hr* obtained during the beginning of the session. The *K_F_* and *ID_F_* are computed using the proposed lightweight permutation algorithm in Section 3.1, where *K* and *ID* are restructured based on the *Hr* value to compute *K_F_* and *ID_F_*. The *Hr* is obtained by the hashing of the *r* value, which is a random number generated at the beginning of each communication session. Thus, *Hr* as well as the computed *K_F_* and *ID_F_* are different for each session. Therefore, attackers are unable to trace any tags from the eavesdropped messages. In addition, the attackers are unable to extract *K_F_* and *ID_F_* from *K_F_*⊕*ID_F_* and, subsequently, are unable to extract *K* and *ID* from the proposed permutation algorithm. 

Attackers might obtain messages *C*, *D*, and *E* by eavesdropping on the communication channel between a RFID tag and a reader. Attackers can then obtain *K_S_*⊕*ID_S_*⊕*m* by XORing messages *C*, *D*, and *E*. Since *m* is a random number freshly generated by the tag for each session, attackers cannot extract *K_S_* and *IDS* from these messages. In addition, all messages *C*, *D*, and *E* are encrypted using random numbers *m* or *s*; thus, attackers cannot perform tracking attacks on RFID tags.

Furthermore, *ID* and *K* are updated at the end of the protocol by XORing random numbers m and s. Note that although *ID*⊕ *K* is equivalent to *ID_new_*⊕*K_new_*, *ID* is not equal to *ID_new_*, and *K* is not equal to K_new_. Since newly generated random numbers and permutated *ID* and *K* values (i.e., *ID_F_*, *K_F_*, *ID_S_*, *K_S_*) are used to compute transmitted messages for each new session, attackers are unable to track a tag because the tag returns no constant response.

## 6. Formal Analysis of the Proposed Protocol

Theoretical analysis and formal analysis tools, rather than experimental analysis, have always been broadly used to analyze security protocols [23]. Thus, in addition to the theoretical security analysis presented in Section 5, the proposed protocol was further analyzed using the formal analysis tool AVISPA. The protocol is written using the High-Level Protocol Specification Language (HLPSL). 

Two back-ends of this AVISPA tool are selected to verify the security of the proposed protocol—On-the-Fly Model-Checker (OFMC) and the Constraint Logic-based Attack Searcher (CL-AtSe) [24]. The other two back-ends are not included in this formal security verification because they are unable to support the exclusive-OR operations used in the proposed protocol. AVISPA uses the Dolev–Yao model for its analysis, where attackers obtain knowledge of normal sessions after its first run. As shown in Figure 5, the OFMC back-end shows that no attack trace was found after searching four nodes in 0.04 *s* with a search depth of 2. CL-AtSe checks whether there is any reachable state wherein attackers might attack and obtain secret keys. If there are reachable states, it analyzes each state to determine whether the safety condition holds or not. The safety condition refers to the situation where attackers are unable to obtain secret keys. The CL-AtSe back-end result shows that no states were reachable to perform security attacks; thus, it implies that it is safe, as indicated in Figure 5. The summary of results of OFMC and CL-AtSe prove that the proposed protocol is secure from replay and man-in-the-middle attacks.

## 7. Performance Analysis

The proposed protocol is compared with the performance of the existing supply chain solution in terms of scalability, transaction cost, interoperability, computational complexity, storage, and security.

### 7.1. Scalability Analysis

Blockchain scalability can be analyzed based on the number of TPS. Currently, Ethereum 2.0 blockchain allows for approximately 36.09 TPS. The scalability of this proposed public blockchain-based supply chain management solution can be improved further by performing batch ownership data transfer associated with RFID tags.

As explained in Section 3.2, the *DATA* file is used for storing secret RFID tag data, and *CID* is generated after uploading the *DATA* file to the IPFS. As transaction input consists of the supply chain node *ID* and encrypted *CID*, this makes the transaction number independent from the number of RFID tags. Therefore, the time needed to perform a transaction does not depend on the number of RFID tags. An experiment was conducted using a Lenovo T14s laptop equipped with an AMD Ryzen™ 5 Pro 4650U central processing unit running at 2.10 GHz base clock speed, with 16 GB of DDR4 random access memory, to analyze the time needed for a transaction to be included in the Ethereum Goerli testnet. For a *DATA* file with RFID data from a random number of tags between 1 and 1000, the transaction time is consistently between 10 and 20 s. The time needed to scan the 1000 RFID tags and update the *DATA* file is not analyzed in this section, as this is outside the scope of public blockchain scalability.

### 7.2. Transaction Fee Analysis

Assuming a supply chain line with 1000 RFID tag-attached goods, a transaction was made with the proposed solution to perform the data transfer of those 1000 tags using the Ethereum Goerli testnet. A total of 292,781 gas was used to execute this *setInput()* function from the *Supply.sol* smart contract. The transaction fee needed was 0.014053488 Ether, which is approximately 16.8 USD at the time of writing. The proposed solution supports the transfer of RFID tags in batches instead of 1000 individual transactions. As a result, this significantly reduces the transaction costs, e.g., by 99% compared to individual transaction costs required for 1000 RFID tags.

### 7.3. Interoperability

The proposed solution provides efficient data management by enabling the sharing of specific data from one supply chain node to another. The proposed solution needs to meet three fundamental privacies: new ownership privacy, old ownership privacy, and a solution to the windowing problem to achieve efficient interoperability. New ownership privacy is preserved by restricting the old owner’s access to the new *DATA* file uploaded by the new owner. This privacy is achieved by encrypting the *DATA* file with the new owner’s public key and then uploading it to IPFS. The proposed protocol also guarantees that the new owner cannot track the previous transactions of the tag because the new owner cannot decrypt the old IPFS *CID* encrypted using another supply chain node’s public key. In order to avoid the windowing problem wherein there should be no time slot for both the old and new owners to access the tag, the new owner should update the *ID* and *K* of the tags once tagged objects are received from the old owner.

### 7.4. Computational Complexity Analysis

Since the proposed system targets resource-constrained IoT devices, the performance of the RFID tags was analyzed to prove that the tags can support the proposed lightweight protocol. According to [1], IoT devices can be categorized intofour classes, mainly depending on their processing capabilities and power consumption. Passive RFID tags are Class I devices, which are resource-limited devices. The total storage cost to store the data that the tags need for the data transfer process is merely the cost of storing *ID*, *K*, and *Hq* values, which amount to 768 bits. This storage size can be supported by passive RFID tags with a chip memory capacity of more than 768 bits. These RFID tag chips include ATA5590 with 1024 bits of user memory and UCODE HSL with 2048 bits of user memory. In addition, Class I devices have low computation capabilities. They cannot support heavy computation outside of simple bitwise operations, such as one-way hash functions and asymmetric encryption supported by Class II devices [4]. The computational cost of an exclusive-OR operation, Txor, is negligible as the cost is less than that for the aforementioned heavy computation [25]. During the authentication process, an RFID tag has a total computational cost of 13 Txor.

### 7.5. Security of Raw Data Storage

The security of raw data storage is vital and can be analyzed in terms of data confidentiality, integrity, and availability. In order to protect data confidentiality, the *DATA* file that stores *ID* and *K* pairs is encrypted using asymmetric encryption before uploading to the IPFS. Although attackers might be able to obtain the encrypted *DATA* file through the *CID*, they are not be able to decrypt the *DATA* file because it can only be decrypted using the private key that was assigned to a specific supply chain node. In addition, the IPFS *CID* is encrypted, and this encrypted string is stored on the Ethereum blockchain to protect its integrity. In order to guarantee data availability upon being requested, all supply chain nodes need to participate as an IPFS node to ensure that at least some IPFS nodes stay online at all times to handle the IPFS process. All supply chain nodes also need to pin the *CID* to ensure important data is retained.

### 7.6. Smart Contract Security Analysis

Smart contracts are immutable. Thus, before deployment, it is vital to ensure that a smart contract is free from vulnerabilities, such as integer overflow/underflow, reentrancy, denial of service, etc. The designed smart contract, Supply.sol, is analyzed using three security tools—Mythx, Slither, and SmartCheck. Supply.sol passed all checks by the aforementioned tools. Mythx is a software-as-a-service platform that provides a higher performance and vulnerability coverage compared to standalone tools such as Slither and SmartCheck. Mythx has three analyzers, where its static analyzer parses the Solidity abstract syntax tree, the symbolic analyzer detects vulnerable states, and the greybox fuzzer detects vulnerable execution paths. Both Slither and SmartCheck belong to static analyzers, which are able to detect simple vulnerabilities faster than Mythx. The details of checks for vulnerabilities covered by these security tools can be found in [26,27,28]. 

Other related solutions do not provide much information on storage and computation cost. However, as shown in Table 5, our proposed system supports Class I IoT devices and outperforms all other systems in terms of security and transaction fees. Furthermore, our proposed protocol allows both batch and solitary data transfer; thus, it provides more flexible and efficient data transfer compared to existing state-of-the-art proposals.

## 8. Conclusions

This paper presented a scalable lightweight protocol for public blockchain-based supply chain systems that uses resource-constrained RFID tags and can transfer RFID tags offline in batches. A lightweight RFID protocol was designed with bitwise exclusive-OR and permutation operations to enable secure communication between RFID readers and tags. A proof of concept was created consisting of a decentralized application deployed on the Ethereum public blockchain and an IPFS for full performance evaluation in a real-world environment. A smart contract was designed and analyzed using formal security tools. The proposed protocol has been proven safe against five attacks using both theoretical and formal analyses. The attacks include those of key disclosure, replay, man-in-the-middle, de-synchronization, and tracking. The proposed lightweight protocol has proven to be efficient in terms of security, transaction cost, scalability, interoperability, storage, and computational cost. Future research will include developing ownership transfer decentralized applications using Non-Fungible Tokens.

## Figures and Tables

**Figure 1 sensors-23-03433-f001:**
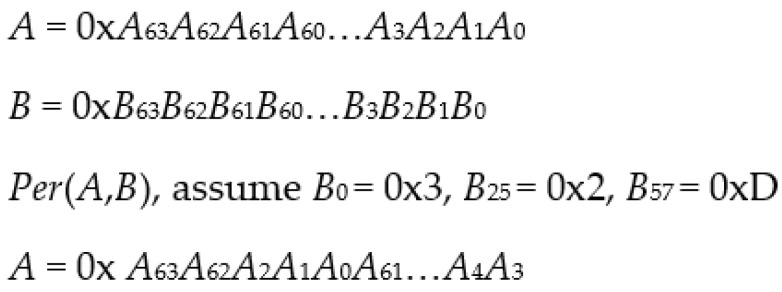
Proposed lightweight permutation operation.

**Figure 3 sensors-23-03433-f003:**
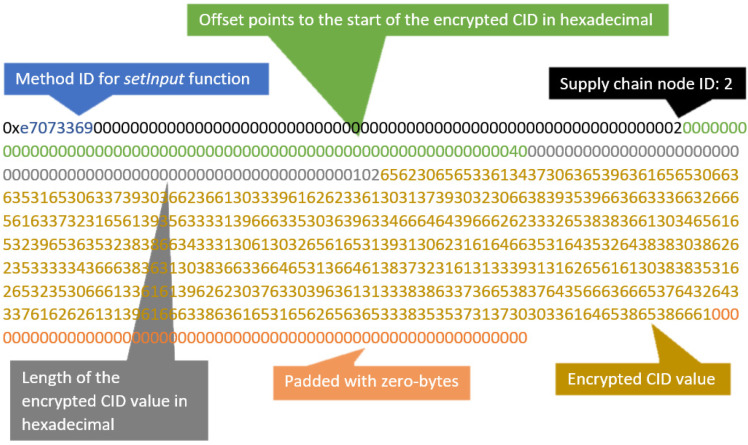
Transaction input.

**Figure 4 sensors-23-03433-f004:**
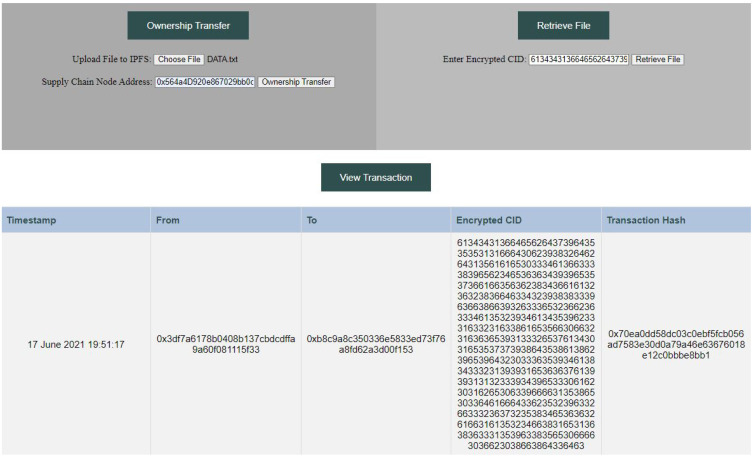
Decentralized web application for RFID tag transfer in a supply chain system.

**Figure 5 sensors-23-03433-f005:**
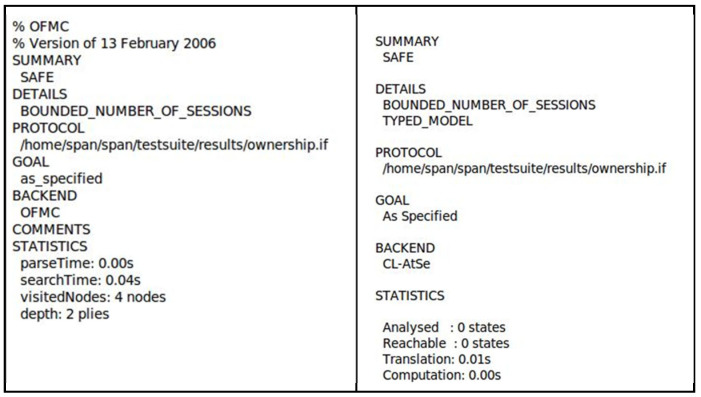
OFMC and CL-AtSe simulation results.

**Table 2 sensors-23-03433-t002:** Permutation operation description.

Description	Specification
*B_0_*	Determines the length of the characters that need to be removed/inserted from/to a string
*B_25_*	Determines the insert position of a string
*B_57_*	Determines the remove and insert direction; the odd number represents characters removed from the left-hand side and inserted at the right-hand side of a string, and vice versa for an even number.

**Table 3 sensors-23-03433-t003:** Supply chain nodes and their functionality.

Entity	Description
Manufacturer	Designing, producing, and delivering products to distributors or retailers
Distributor	Distributing products purchased from manufacturers to retailers
Retailer	Selling products purchased from distributors to end-users
End-user	Purchasing products from retailers

**Table 4 sensors-23-03433-t004:** Notations used in the proposed protocol.

Notation	Description
⊕	Exclusive-OR
*m*	A random number generated by a tag
*r*	A random number generated by a reader
*s*	Random number generated by a supply chain node
*n*	Random number generated by a supply chain node
*q*	Random number generated by a supply chain node
*Hr*	Hash string generated from *r* random number
*ID*	Current session pseudonym
*ID_new_*	Next session pseudonym
*ID_F_*	A permutated session pseudonym computed by a tag
*ID_S_*	A permutated session pseudonym computed by a supply chain node
*K*	Current session secret key
*K_new_*	Next session secret key
*K_F_*	A permutated session secret key computed by a tag
*K_S_*	A permutated session secret key computed by a supply chain node
*Qm*	Current owner’s IPFS content hash string
*Qm_new_*	New owner’s IPFS content hash string
*H*	Current owner’s encrypted IPFS content hash string
*H_new_*	New owner’s encrypted IPFS content hash string
*Hq*	Current hash string generated from *q* and *n*
*Hq_new_*	New hash string generated from *q* and *r*
*Tx*	Blockchain transaction hash string
*hash*	SHA-256
*||*	Concatenation
*′*	A message computed using one or several received messages

**Table 5 sensors-23-03433-t005:** State-of-the-art and proposed protocol comparison.

Description	Proposed System	VeChain [5]	Waltonchain [6]	Rozman et al. [9]	Ahmad et al. [15]
Security protection from					
Key disclosure	Yes	Yes	Yes	NA	No
Replay	Yes	Yes	Yes	NA	No
MITM	Yes	Yes	Yes	NA	Yes
De-Sync	Yes	Yes	Yes	NA	Yes
Tracking	Yes	No	No	NA	NA
Performance					
Secure smart contract	Yes *^1^	Unknown	Unknown	Unknown	Yes
Storage provider	IPFS	CHAOS	Child chains	Side chain	IPFS
IoT device class support	I	II	II	III, IV	Unknown
Transaction fees (USD)	16.80 *^2^	27.00 *^2^	Variable	Variable	Variable
Batch transfer	Yes	No	No	No	No
Scalability	Yes	No	No	Yes	No
Interoperability	Yes	No	No	Yes	Yes

*^1^—Analysis performed using Mythx, Slither, and SmartCheck. *^2^—Transaction cost associated with ownership transfer of 1000 RFID tags.

## Data Availability

Relevant data are contained within the article.

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
