# Peer review of "Scalable Lightweight Protocol for Interoperable Public Blockchain-Based Supply Chain Ownership Management"

_sensors, 2023, doi:10.3390/s23073433_

Round 1

Reviewer 1 Report

  1. Check all the captions of the Table. All captions must be of the same style.
  2. The term IPFS is not defined in the contribution list, so define the abbreviation on its first occurrence.
  3. "In this paper, the Ethereum blockchain is proposed to be used" revise this sentence.
  4. The authors need to elaborate the figure 2, like what is supply chain node and its function. In addition, the authors need to explain the other entities in figure 2 for the smooth reading of the paper.
  5. What is "manufacturer node" in the registration phase?
  6. How the integrity of the communicated messages is checked in the proposed scheme.
  7. How the proposed scheme is resistant to the impersonation attack.
  8. Is the proposed scheme resistant to the device capture attack?

Author Response

Dear Reviewer, 

Thank you for very much for your feedback. We have made corrections to the manuscript and are attaching our response in a letter. Please refer to the attachment.

Sincerely yours

Reviewer 2 Report

1. in related work part, good summary is required.

2. in 3.1 “Per”,in 3.2 “PRE”, are they the same ?

3. in 3.1, Per(A,B) is not well defined, an explanation of Per is needed. B57=D, what is the meaning of D?

4. in line 168, “two generated random numbers,n,q” and  in table 3 "s,n,q" are the same meaning?

5. Figure 4 is unclear, please make sharp figure for it.

Author Response

(The authors gave the same response as above.)
